Mechano-chemical regulation of complex cell
shape formation: Epidermal pavement cells—A
case study. *Quantitative Plant Biology*, **4:**e5,
1–10 https://dx.doi.org/10.1017/qpb.2023.4

cell shape; cell wall; mechanical stress;
microtubules; morphogenesis; pavement cells.

**Corresponding author:**
Arun Sampathkumar;
Email: sampathkumar@mpimp-golm.mpg.de

# Mechano-chemical regulation of complex cell shape formation: Epidermal pavement cells—A case study

Ruben van Spoordonk[1], René Schneider[2] and Arun Sampathkumar[1] 

[1]Max Planck Institute of Molecular Plant Physiology, Potsdam, Germany; [2]Institute of Biochemistry and Biology, Plant Physiology Department, University of Potsdam, Potsdam, Germany

## Abstract

All plant cells are encased by walls, which provide structural support and control their morphology. How plant cells regulate the deposition of the wall to generate complex shapes is a topic of ongoing research. Scientists have identified several model systems, the epidermal pavement cells of cotyledons and leaves being an ideal platform to study the formation of complex cell shapes. These cells indeed grow alternating protrusions and indentations resulting in jigsaw puzzle cell shapes. How and why these cells adopt such shapes has shown to be a challenging problem to solve, notably because it involves the integration of molecular and mechanical regulation together with cytoskeletal dynamics and cell wall modifications. In this review, we highlight some recent progress focusing on how these processes may be integrated at the cellular level along with recent quantitative morphometric approaches.

## 1. Introduction

How subcellular processes influence cell shape and thus contribute to the overall architecture of organisms is a central question in biology. In this review, we provide an overview of the molecular and mechanical factors contributing to the morphogenesis of the complex jigsaw puzzle-shaped morphology of epidermal pavement cells found in leaf-like tissues. While we focus on more recent findings in addition to highlighting some of the seminal work on the subject, in-depth reviews on the topic should be referred to for more detailed insights on pavement cell morphogenesis and plant biomechanics (Liu et al., 2021; Sampathkumar, 2020; Trinh et al., 2021).

## 2. Biochemical regulation of pavement cell morphogenesis

### 2.1. Spatial control of ROP GTPase during pavement cell morphogenesis

Rho guanosine triphosphate (GTP)ases are a family of membrane-bound, small signalling G-proteins that are known regulators of the cytoskeleton in eukaryotes. In plants, there is only one family of Rho GTPases referred to as 'Rho of plants' (ROP). Members of the ROP family have been identified as important signalling molecules in many cellular responses and developmental processes. They are known to do so often by interaction with members of the 'ROP-interactive CRIB motif-containing' (RIC) protein family (Fu et al., 2005, 2009; Lin et al., 2015; Xu et al., 2010). In pavement cells, two specific ROP-RIC pairs (ROP2/ROP4-RIC4 and ROP6-RIC1) have been shown to respectively govern the formation of alternating protrusions and indentations (Figure 1; Fu et al., 2005; 2009; Lin et al., 2015; Xu et al., 2010; 2011). These ROPs are thought to be activated in alternating domains in the plasma membrane along the anticlinal face (i.e., perpendicular to the leaf surface). Each ROP then sets in motion a separate downstream signalling pathway through interaction with their respective RIC. ROP2 and ROP4 act redundantly and interact with and activate RIC4, which then induces the local assembly of F-actin at the plasma membrane. This was then proposed to promote/facilitate exocytosis, thereby enabling local cell outgrowth resulting in the formation of a protrusion. ROP6, on the other hand, activates RIC1,

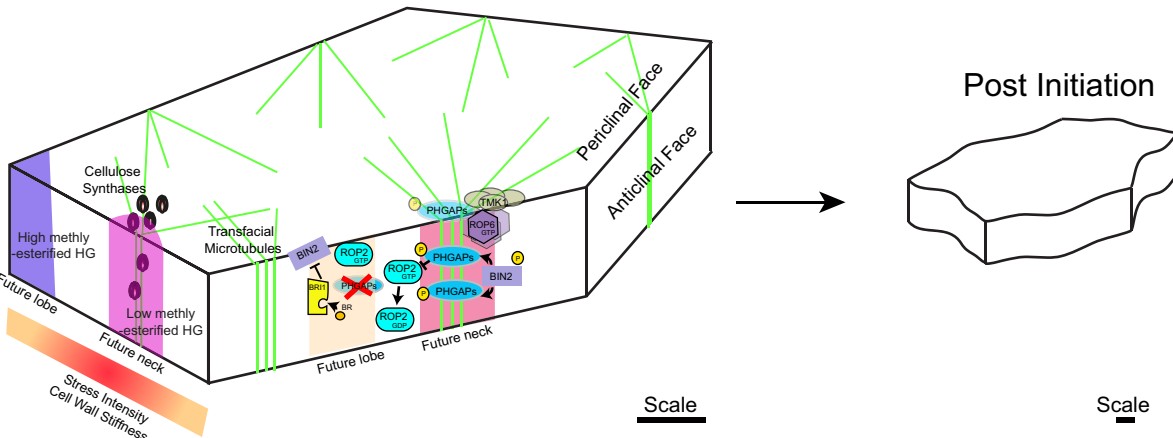

**Figure 1.** Process of symmetry breaking in pavement cells. The illustration on the left-hand side highlights cell wall based modific ation and ROP signaling pathways that initiate the formation indentations (neck) and protrusions (lobe) in pavement cells.

which then goes on to deploy the microtubule-severing protein KATANIN (KTN). KTN activity promotes the local assembly of microtubules, which results in the deposition of stiff cellulose microfibrils (CMFs) in similar regions of the anticlinal wall, leading to the formation of an indentation by locally suppressing outgrowth. These two pathways, ROP2/ROP4-RIC4 and ROP6-RIC1, have also been shown to laterally inhibit each other, reinforcing the alternated nature of their domains (Fu et al., 2005; 2009; Lin et al., 2015). ROP2-RIC4 signalling inhibits RIC1 activity preventing microtubule assembly in ROP2-governed domains. At the same time, the presence of microtubules in ROP6-governed regions inhibits RIC4 activity in those domains preventing F-actin assembly. In this way, the alternation of protrusions and indentations is reinforced and maintained. However, it is yet unknown how the microtubules cause inhibition of RIC4 activity, although it seems likely to involve some microtubule-associated proteins. Interestingly, Sugiyama et al. (2017) report that an IQ-domain protein (IQD13) fulfils such a role during xylem formation. During xylem formation, cells produce thick secondary cell walls except in the secondary wall pits, which form lateral connections between mature xylem vessels (Oda & Fukuda, 2012a). The formation of these pits is governed by ROP11 which counteracts microtubule assembly in those domains through a pathway with MICROTUBULE DEPLETION DOMAIN1 (MIDD1) and KINESIN-13A (Oda & Fukuda, 2012b, 2013; Sugiyama et al., 2017). Sugiyama et al. (2017) found that IQD13 interacts with both the plasma membrane and microtubules. It promotes microtubule rescue and inhibits ROP11 activity. In this way, IQD13 restricts ROP11 activity to its confined domains (Sugiyama et al., 2017). Whether IQD13 or a related protein would fulfil such a role in pavement cell morphogenesis formation remains to be investigated.

ROPs can switch between an active GTP-bound form, present in the plasma membrane, and an inactive guanosine diphosphate (GDP)-bound form, detached from the plasma membrane based on the binding of its regulators. GUANINE NUCLEOTIDE EXCHANGE FACTORS (GEFs) facilitate the exchange of the GDP to GTP promoting the binding of the active ROPs to its downstream effectors (Hodge & Ridley, 2016), whereas GTPase ACTIVATING PROTEINs (GAPs) boost the hydrolysis of the bound GTP rendering the ROPs inactive. In Arabidopsis, two groups of ROPGAPs are present: one that contains the Cdc42/Rac-interactive

binding (CRIB) domain and another group that contains an amino-terminal pleckstrin homology, instead of the CRIB, domain called PHGAPs (Hwang et al., 2008). Recently, PHGAPs were found to accumulate on the anticlinal face of the cell along indentations and inactivate ROP2, resulting in differential activation of ROPs necessary for the formation of the puzzle-piece morphology of pavement cells (Figure 1; Lauster et al., 2022).

### 2.2. Hormonal control of pavement cell morphogenesis

Phytohormones are also associated with pavement cell development via their regulation of ROPs. Brassinosteroids, a group of steroid hormones, have been shown to influence pavement cell shape by stabilising microtubules by a negative regulation of a brassinosteroid response GSK3-like kinase BR-INSENSITIVE2 (BIN2) (Liu et al., 2018). More recently, BIN2-based phosphory-lation of PHGAPs was shown to stabilise PHGAP presence in the indenting domains influencing ROP2 function (Figure 1; Zhang et al., 2022). Cytokinins, another group of plant hormones, were also shown to act upstream of the ROPs, with both signalling and biosynthetic mutants showing changes in pavement cell shape (Li et al., 2013).

Auxin signalling is well studied and proposed to be involved in the formation of puzzle-piece shapes and over the past decade there has been progress in understanding the signalling mechanisms (Belteton et al., 2018; Grones et al., 2020; Nagawa et al., 2012; Pan et al., 2020; Platre et al., 2019; Xu et al., 2010). It has been shown before that auxin triggers ROP activation, as is true for ROP2 and ROP6, and that this is mediated by a complex of AUXIN BINDING PROTEIN 1 (ABP1) and TRANSMEMBRANE KINASE 1 (TMK1, Lin et al., 2015; Xu et al., 2010; 2011; 2014). In this complex, ABP1 acts as the extracellular auxin receptor where TMK1 transfers the signal to the cell's interior. In fact, an earlier review by Lin et al. (2015) already suggested a model where this signalling complex is integrated with the ROP-RIC pathways to explain the molecular regulation of pavement cell lobe and neck formation. This model may now be expanded upon by introducing the process of nan-oclustering. Several components already in the model have been observed to form auxin-induced nanoclusters at the plasma mem-brane. In one study, auxin presence resulted in reduced diffusion rates of TMK1 particles promoting the formation of nanoclusters

in future indentation domains of pavement cells (Pan et al., 2020). These TMK1 nanoclusters signal clustering and activation of ROP6 at the plasma membrane (the precise mechanism of which remains yet unclear) resulting in cortical microtubule changes (Figure 1; Pan et al., 2020). The microtubules would then be involved in stabilising these TMK1 nanoclusters closing a positive feedback loop for formation of indented domains (Pan et al., 2020). This is supported by an earlier study where auxin signalling triggers the nanoclustering of ROP6 on the plasma membrane of root cells. Here the process is shown to be mediated by the nanoclustering of phosphatidylserine (Platre et al., 2019), indicating a possible role for specific plasma membrane components.

Further evidence for auxin involvement can be found when studying auxin transport. Indeed, many reports show auxin transporter proteins as a necessary component in pavement cell morphogenesis (reviewed in Liu et al., 2021). Specifically, PIN-FORMED (PIN) auxin transporters have been indicated in relation to the ROP signalling pathways (Nagawa et al., 2012). In detail, ROP2-RIC4 signalling is shown to promote PIN localisation at the (future) protrusion regions through local inhibition of endocytosis (Nagawa et al., 2012), which should increase auxin efflux at the tip of the protrusion. The extracellular auxin may then bind locally to ABP1 on the protrusion surface as well as on the indenting domains of the neighbouring cell. This may then trigger ROP2 signalling in these protrusions, creating positive feedback, and at the same time trigger ROP6 signalling in the indentations of the neighbouring cell. This model provides a mode of communication between neighbouring cells to coordinate the placement of their protrusions and indentations. However, in this case, it still seems to be dependent on the alternating ROP domains.

Several studies show a functional role of auxin during pavement cell development. However, automated detection of protrusions in pavement cells found no significant changes in their number or overall cell shape in a range of PIN mutants (Belteton et al., 2018). Grones et al. (2020) took advantage of a subpopulation of leaf epidermal cells known as the stomatal lineage ground cells (SLGCs) generated by stomatal meristemoids to re-investigate the role of auxin in formation of puzzle-shaped cells in a time-resolved fashion. SLGCs are generated by asymmetric cell division in a spiralling fashion around the meristemoid cell, together forming an anisocytic spiral complex. The SLGCs are initially polygonal and an ascending auxin response gradient influences the generation of indentations (Grones et al., 2020). In an early stage of anisocytic spiral complexes, these gradients were predominantly ascending from the centre outward, showing the lowest auxin response in the meristemoid cell and the highest in the oldest SLGC of the complex (Grones et al., 2020). The formation of these gradients was shown to be dependent on auxin transport, rather than auxin biosynthesis, and involves several different auxin transporters like PINs, AUX1, and ABCB (Grones et al., 2020). Mutations of these transporters lead to cell size and cell shape defects in pavement cells of true leaves but not in cotyledons (Grones et al., 2020). This implies that regulatory mechanisms of morphogenesis may differ between pavement cells of cotyledons and of true leaves (Grones et al., 2020).

## 3. Mechanics behind pavement cell morphogenesis

### 3.1. The plant cell wall

The local mechanical properties of the cell wall are a major determinant of local cell growth and thus directly affect the overall morphology of the plant. The differential distribution of microtubules

mentioned above has consequences in how the stiffest component of the cell wall, the CMFs, is deposited. Cellulose is synthesised directly at the plasma membrane by membrane-spanning cellulose synthase complexes (CSCs), which are transported to the plasma membrane via the Golgi and trans-Golgi network, where they are delivered in proximity to microtubules (McFarlane et al., 2014). Within the plasma membrane, CSCs move at a constant speed. However, their direction of movement is governed by the direction of cortical microtubules to which the CSCs are coupled by CELLULOSE SYNTHASE INTERACTING 1 (CSI1/POM2) proteins (Bringmann et al., 2012; Paredez et al., 2006). Ongoing synthesis of new CMFs continuously extrudes stiff cellulose fibres into areas with older CMFs, causing mechanical tension to build up within them that is relieved by the displacement of CSCs within the plasma membrane along cortical microtubules (Diotallevi & Mulder, 2007; Morgan et al., 2013). Partial disruption of microtubules leads to changes in CSC trajectories at the plasma membrane, and complete removal of microtubules by the drug oryzalin leads to the uniform movement of CSCs at the plasma membrane. Although cortical microtubules are required for the directional movement of CSCs at the plasma membrane, they do not affect CSCs motility or cellulose synthesis (Sugimoto et al., 2003). Unlike cellulose, other cell wall matrix components are pre-synthesised in the Golgi apparatus, transported to the plasma membrane and extruded into the cell wall (Lerouxel et al., 2006).

CMF–CMF interactions are widely present in the cell wall and in such regions, the presence of hemicellulose xyloglucan is also observed. It is proposed that the activity of expansion on those sites might mediate the remodelling of the nanostructure of the cell wall, thereby influencing the mechanics locally (Cosgrove, 2014).

### 3.2. Microtubule and cellulose-based models on pavement cell morphogenesis

Long-term kinematic imaging of microtubules in developing pavement cells combined with microtubule de-polymerisation experiments suggest that microtubules play an important role in shape emergence (Armour et al., 2015), whereas other independent studies suggest that microtubules are not consistently present at prospective sites of invaginations along the anticlinal walls (Belteton et al., 2018). A two-step mechanism in which pectin-mediated changes to wall mechanics occur first and initiate the symmetry-breaking event followed by the presence of microtubules in such domains enhancing the morphology by guiding cellulose synthesis along similar directions (Altartouri et al., 2019). Studies until now proposed a growth restriction-based model of pavement cell morphogenesis, where anticlinal microtubules mediate growth restriction in the indentations and actin mediates growth promotion in the protrusions (Sapala et al., 2018). However, these studies lacked detailed monitoring of growth behaviour at the subcellular scale. Tracking of fluorescently labelled plasmodesmata along the cell-to-cell interface of developing pavement cells disputes growth restriction-based models. The results of the study showed no differences in rates of expansion in subdomains of both indentations and protrusions measured based on the displacement of existing plasmodesmata. The authors of the same work propose that microtubules that transition from the periclinal to the anticlinal face (transfacial microtubules) could promote the initiation of indentations by regulating CMF deposition in such domains (Belteton et al., 2021). Such patterns of CMF deposition could restrict expansion of cell height promoting expansion along the plane parallel to the leaf face facilitating morphogenesis.

### 3.3. Mechanical stress-based models on pavement cell morphogenesis

Atomic force microscopy-based measurements of the cell wall stiffness causally link the patterns of molecular effectors (microtubules) and differences in wall mechanics (Sampathkumar et al., 2014). Regions of stable microtubule presence correlate with stiffer domains of the cell wall. Finite element model of these cellular geometric features in combination with turgor pressure shows increased magnitude and highly anisotropic mechanical tensile stresses (herein referred to as mechanical stresses) being present in the indenting regions (Bidhendi et al., 2019; Sampathkumar et al., 2014; Sapala et al., 2018). In addition, micromechanical manipulation of existing stresses results in concomitant changes to microtubule organisation along the newly predicted direction of stress (Sampathkumar et al., 2014). These results demonstrate the existence of a mechanical feedback loop where cell geometry-driven subcellular stresses regulate microtubule organisation that promotes stiffening of indented regions resulting in the maintenance of cell shape (Chebli et al., 2021; Jonsson et al., 2022; Sampathkumar, 2020; Trinh et al., 2021). More recently, the transfacial alignment of the CMFs from the periclinal to the anticlinal wall was proposed to generate mechanical stresses hotspots that promote microtubule – cellulose-mediated modification to the mechanics of the cell wall that promotes the initiation of the puzzle-shaped morphology (Figure 1; Belteton et al., 2021; Jonsson et al., 2022). Such complex morphological features are shown to be necessary to reduce mechanical stresses in cells that attain large volumes while growing in an isotropic fashion (Sapala et al., 2018).

### 3.4. Pectin-based models on pavement cell morphogenesis

Apart from cellulose and hemicellulose, pectin is an important cross-linking component of the cell wall that influences its stiffness. The major component of pectin is homogalacturonan and the esterification status of which determines the cell wall stiffness. This is regulated by the activity of PECTIN METHYLESTERASE and its inhibitor PECTIN METHYLESTERASE INHIBITOR. Finite element models suggest that initiation of indentations occurs due to mechanical heterogeneities present along the anticlinal wall between two adjacent cells (Figure 1; Majda et al., 2017). Such mechanical heterogeneities are proposed to be created by the polar distribution of demethylated pectin and other cell wall components such as galactans and arabinans based on immunolabelling experiments. Presence of both esterified and non-esterified pectin was detected at the interface of the periclinal and anticlinal walls of protrusions in some angiosperm and fern species (Sotiriou et al., 2018). This potentially indicates a high amount of turnover and remodelling of pectin in the protruding domains that could contribute to shape changes. Super resolution-based imaging indicates that pectin exists as nanofilaments along the anticlinal wall in pavement cells (Haas et al., 2020). Further, an increase in pectin nanofilament width and spacing occurs due to the demethylation of the pectin by PMEs. The same study used finite element models to propose that the indentations of straight anticlinal walls occur due to changes in the ultra-structure of the pectin nanofilaments independent of turgor pressure or CMF-based reinforcements. However, such nanofilaments are absent in the periclinal wall (i.e., parallel to the cell surface) and both models (Haas et al., 2020; Majda et al., 2017) fail to consider the mechanics of the periclinal wall. It was also shown that the addition of the periclinal cell wall eliminated minor cell wall bending occurring along the anticlinal cell wall (Bidhendi

& Geitmann, 2019). The presence of demethylated pectin correlated with stiff domains of the periclinal cell wall at prospective sites of indentation further highlighting the importance of periclinal wall mechanics (Figure 1; Bidhendi et al., 2019). The study also proposes that pectin-mediated shape changes generate stress differentials that attract microtubules to such sites further enhancing the morphological features (Bidhendi et al., 2019). However, it is still unknown what facilitates the heterogeneous presence/activity of the pectin modifying enzymes.

## 4. Mechanical stress-based microtubule response is subject to mechanical noise

Studies clearly indicate that microtubules are under the control of mechanical stresses, yet it is unclear how microtubules sense mechanical stresses, while there are several proposed biochemical pathways that could potentially contribute to mediating a microtubule response towards mechanical signals (Trinh et al., 2021). Microtubule responses to changes in mechanical stresses remain to be validated in majority of these mutants. Most of the mutants studied so far that have defects in their microtubule response to stress are also central regulators of microtubule dynamics, thereby most likely contributing to the mechano-response rather than serving as mechanical sensors (Eng et al., 2021; Hervieux et al., 2016; Sampathkumar et al., 2014; Takatani et al., 2020; Uyttewaal et al., 2012). This includes the microtubule-severing enzyme KTN which is upregulated upon changes to mechanical stress (Sampathkumar et al., 2014). It was recently proposed that pectin interaction with a receptor kinase FERONIA recruits GEF14 which activates downstream ROP6 facilitating the microtubule response to mechanical stress (Tang et al., 2022). However, an earlier study on a mutant of *feronia* showed opposing results on its putative role as a mechanosensor (Malivert et al., 2021). Therefore, the involvement of receptor kinases in acting as mechanosensors in plants remains to be further evaluated. Another possibility is the hypothesis that microtubules themselves serve as sensors of mechanical stress, with curvature as the instructional cue (Hamant et al., 2019). Quantitative assessment of microtubules in relation to curvature in pavement cells, however, shows only a moderate correlation between microtubules and the predicted mechanical stress, indicating that microtubule response to mechanical stress is to a certain extent noisy (Eng et al., 2021; Schneider et al., 2022). Studies indicate that the activity of a few microtubule regulatory proteins such as NEK6, SPIRAL2 and CLASP does indeed contribute to the dampening of microtubule response to mechanical stress (Eng et al., 2021; Hervieux et al., 2016; Takatani et al., 2020). Such differences in microtubules also impact how CSC presence corresponds to mechanical stress (Schneider et al., 2022). Only a moderate correlation of microtubules with the predicted mechanical stress (based on 2D curvature of anticlinal walls) was found in developing pavement cells (Eng et al., 2021). The highest correlation occurs during phases of rapid growth, presumably due to increases in the internal turgor pressure. A similar observation of relatively weak correlation was observed between curvature and CSC presence, with no differences in velocity of CSC movement between regions of high versus low mechanical stress in pavement cells (Schneider et al., 2022). The study also demonstrated that such reductions in correlation between microtubule organisation and curvature occur due to the interaction of the microtubules to CMF via CSCs and CSI1/POM2 that prevents microtubules to move freely along certain paths. Supportive of this, microtubule presence in the *csi1* mutant better

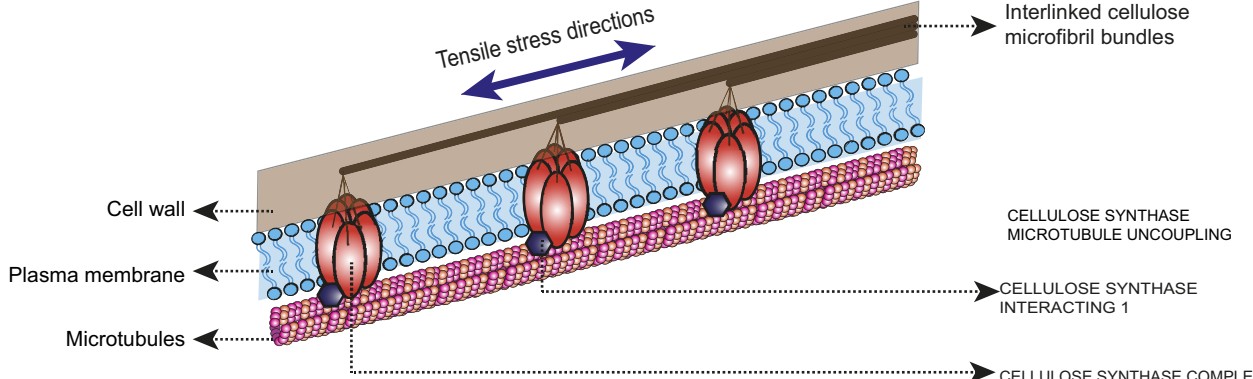

**Figure 2.** Biophysical continuum. A graphical representation of the coupling of the cellulose synthase complexes to the nascent cellulose microfibrils in the cell wall and the intracellular microtubules via associated proteins. Cellulose microfibril deposition occurs along the principal direction of mechanical stress assisted by the microtubule cytoskeleton. This physical continuum adds to disturbances in microtubule organization and contributes to transduction of mechanical forces from the cell exterior (cell wall) to the cells interior (microtubules).

correlates with regions of negative curvature as their association with wall CMF is reduced. Further, changes to mechanical stress resulted in the depletion of CSCs from the plasma membrane, potentially facilitating the microtubule response. Collectively, these results indicate that the physical continuum observed between CMF and microtubules contributes to noisier microtubule organisation and serves as a medium for the transduction of mechanical stresses (Figure 2). It remains to be tested how such differences in microtubule organisation contribute to the adaptive response of microtubules to internal and external cues.

## 5. Quantitative approaches to evaluate pavement cell morphology

Although morphogenesis is a fascinating subject of plant research for over 100 years (Thompson, 1917), the interest of plant researchers in the formation of shapes has increased significantly in recent years. Regardless of whether the shapes of organs, ranging from ovules (Vijayan et al., 2021) to leaves (Bhatia et al., 2021; Kierzkowski et al., 2019; Zhao et al., 2020) and flowers (Rebocho et al., 2017), or isolated tissues such as leaf pavement cells and the shoot apical meristem (Hamant et al., 2008; Sampathkumar et al., 2019) or even unicellular models such as trichomes and root hairs are the matter of study, dynamic recording and quantification of growth processes are applied in almost all areas, which has given new impetus to the field of morphodynamics. In this context, the abundance of data and the ease of recording (by today) are rapidly saturating existing methods for describing and quantifying shapes. Below we review a selection of quantification tools and parameters that have invigorated and found application in the plant research community in recent years.

Confocal fluorescence microscopy has become the method of choice when studying the cellular dimensions of plant tissues. For most morphodynamic studies, cells of the epidermal layer are studied and Z-stacks of membrane-localised markers such as LTi6B (Cutler et al., 2000) and myrYFP (Willis et al., 2016) or membrane-specific dyes such as FM4-64 or dyes that label cell wall like propidium iodide are recorded. For this purpose, the cells are often embedded in special imaging chambers to follow morphogenesis over a period of hours to several days under near-physiological conditions (Seerangan et al., 2020).

To determine the shape of the cells, a single plane is often selected from the Z-stack. This provides a cross-section through the cells of interest allowing extraction of the cell outlines (i.e., generation of a list of x-y coordinates of boundary pixels) using automatic, and less commonly manual, segmentation tools. This approach largely ignores depth information and thus provides a simplified, but in some cases sufficient, representation. Other approaches use projection methods where the three-dimensional shape of the cells is considered. This is done either by creating entire 3D models that are computationally more complex or by creating a surface mesh that represents the original cell shape as a curved plane, often referred to as a 2.5D model (Barbier de Reuille et al., 2015; Eng et al., 2021; Erguvan et al., 2019; Haertter et al., 2022; Herbert et al., 2021; Schneider et al., 2022).

Which parameters are suitable for quantifying cell shape? The simplest approaches include ratio approaches such as the aspect ratio, which relates the longest to the shortest cell axis, or circularity, which relates the circumference of an object to its area (see Figure 3 for definition). In both cases, more circular objects lead to values close to '1'. The more the cell shape deviates from a circle, for example, by deformation or growth along a preferred axis, the more the values converge towards '0'. Both parameters thus express how isotropic or anisotropic an object is but can hardly make reliable statements about the complexity of the cell shape. Other measures have proven more reliable in this regard, for example, solidity, which relates the area of the cell to the area of its convex hull. The convex hull describes the shortest path around a cell. Similarly, lobeyness relates the length of the convex hull and the perimeter of the cell. Again, both parameters are constructed to give '1' for perfect circles and smaller values for objects with indentations (see Figure 3). Thus, both methods can capture geometric features that have 'negative' or concave curvature. Since morphogenesis can be understood as the process that generates concave curvatures in biological systems, both parameters provide reliable albeit simplistic information. Caution is therefore required in interpretation, as these methods cannot distinguish between cells with few but large protrusions and cells with many but small protrusions.

The emergence of indentations and protrusions of the cell contour represents an interesting biological phenomenon. Quantification of such features has been an important area of research and method development in recent years (Möller et al., 2017; Nowak et al., 2021; Sánchez-Corrales et al., 2018; Wu et al., 2016). In this context, the number and dimensions of indentations and protrusions have been used in various approaches for screening mutants. Less common is the use of curvature to

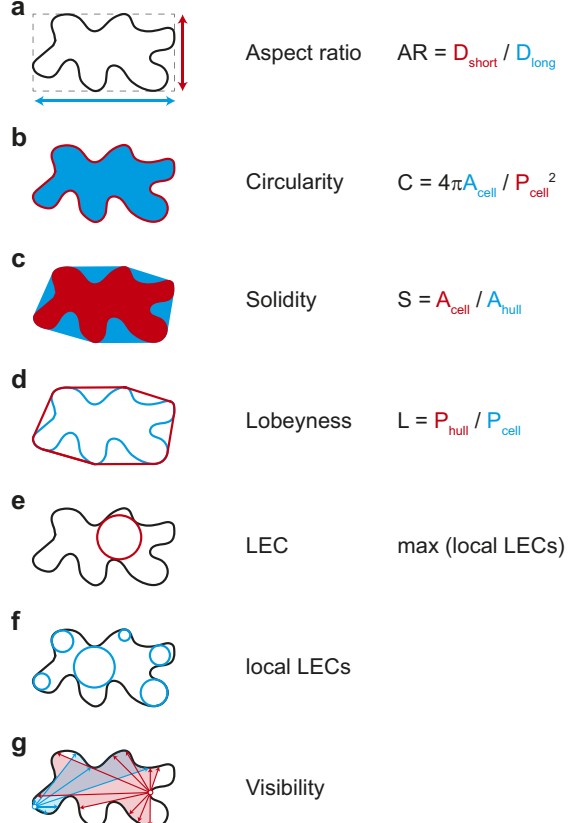

**Figure 3.** Selection of geometric parameters used to quantify cell-shape. (A-D) Global parameters use geometric features of the entire cell: longest and shortest axis in aspect ratio (A), cell area and perimeter in circularity (B), cell area and convex hull area in solidity (C), and cell perimeter and convex hull perimeter in lobeyness (D). Local parameters use local features of the cell outline: the largest empty circle (LEC) is defined as the largest circle that just fits into the 2D cell outline (E) whereas the local LECs (F) are computed by placing at each point along the contour the largest possible empty circle that just fits within the outline of the 2D cell. Network properties, such as visibility (G), are derived from the position of contour points in relation to a query point (red and blue lines).

describe phenotypes. This has nevertheless been used recently to link information about changes in cell shape to the localisation of microtubules and cellulose synthases within cells (Eng et al., 2021; Schneider et al., 2022). Another parameter that is increasingly finding its way into morphodynamic studies is the 'largest empty circle' (LEC) (Sapala et al., 2018). This parameter describes the largest circular area that just fits within a cell. It has been shown that the mechanical stress on the cell wall is greatly increased in such areas (Eng et al., 2021; Sapala et al., 2018). Thus, the LEC can be used as an indicator of maximal cell wall stress and serve as a link between cell shape and mechanical forces (Eng et al., 2021). These parameters are explained schematically in Figure 3. Methods have been developed that extract those features semi-automatically based on different mathematical entities: curvature, envelope and elliptical Fourier analysis. More recently, the network properties of the image pixels have been exploited, for example, by decomposing the cell contour into individual points (nodes) and representing their position relative to each other (edges) as a network graph. One result of this is 'visibility', which measures how many points of the contour can be connected to a selected point of the contour in a straight line. In the following, we are providing a summary of these methods.

## 5.1. PaCeQuant: Curvature-based method (Möller et al.,2017)

PaCeQuant is an ImageJ-based plugin that measures many geometric and biologically interpretable parameters of projected Z-stacks (images). These include (a) cell features such as area, perimeter and circularity, (b) contour-based features such as local curvature on which indentation and protrusion detection is based and (c) skeleton-based features such as longest path and number of branches, and finally cell-specific features such as the number of lobes, lobe length and lobe width. The tool includes the option to perform segmentation, which allows the user to start directly with confocal images. Due to the computationally efficient quantification of many features, large datasets can be analysed relatively quickly with this tool. The drawback is that each cell is analysed on its own, and thus, no reference can be made between neighbouring cells and intracellular entities (e.g., microtubules or other proteins). Furthermore, the tool can distinguish between true lobes versus lobes associated with tri-cellular junctions, but their position is not output directly.

## 5.2. Contour analysis: A curvature-based method (Eng et al., 2021; Schneider et al.,2022)

This is a MATLAB-based tool that uses 2.5D projections of Z-stacks (images). However, multiple colour channels can be used in parallel, which additionally contain information about fluorescently labelled proteins. The tool performs watershed segmentation to first extract digitised cell outlines. It is one of the few tools that then correlate the abundance of fluorescently labelled proteins near the inner cell boundary with the local curvature. Curvature is determined by fitting a circle to the local cell boundary. Local curvature and mechanical strain are closely related, as confirmed by Eng et al. (2021). Therefore, it is worthwhile to use a modified, local version of the LEC (localLEC), which determines for each point along the contour the largest possible circle that just fits inside the cell at that location (Figure 3e,f). Also incorporated into this tool is the FibrilTool algorithm (Boudaoud et al., 2014) that extracts anisotropy and orientation of fibrillar structures both locally in relation to curvature and globally over the entire cell surface. The contour analysis tool can be downloaded from GitHub (https://github.com/DrReneSchneider/Matlab-Contour-Analysis). An optional 2.5D projection method developed by the same team of researchers that applies a smooth manifold algorithm to 3D Z-stacks to extract fluorescent signals from the cell surface can be downloaded also from GitHub (https://github.com/DrRene Schneider/Smooth-Manifold-Projection-Tool) and used as a preprocessing step prior to contour analysis. As with the other techniques, this tool does not directly provide the positions of lobes, necks and tri-cellular junctions. As the cell contour is analysed in terms of curvature, a clear distinction between lobes and tri-cellular junctions is problematic. However, the ability to sort out cell walls adjacent to stomata is useful.

## 5.3. LobeFinder: Envelope-based method (Wu et al.,2016)

LobeFinder is a special application that aims to measure the complexity of cells based on the number of indentations and protrusions. It uses projected Z-stacks (images) and requires manual segmentation of cell outlines. The method is based on the determination of a convex envelope (hull) for 2D cell outlines (Figure 3). Since the envelope runs along protrusions but cannot follow the indentations, the distance between the envelope to the actual cell

boundary can be used as a position indicator for these features. Although this method has been used in time-lapse imaging, it is not particularly sensitive to dynamic changes in the cell contour. Thus, an emerging curvature is likely to be overlooked due to the initially small distance to the envelope and detected only when the change is already substantial. Nevertheless, the method is a robust tool to describe the emergence/increase of complexity during growth and between mutants. Again, the problem arises that these features (indentations and protrusions) are analysed individually for each cell and no correlation can be established either to the neighbouring cells or to the cellular content (e.g., microtubules). The possibility of misinterpretation is therefore considerable.

### 5.4. GraVis: A network-based method (Nowak et al.,2021)

This method employs ideas stemming from network theory, namely visibility graphs and closeness centrality to extract, quantify and compare shape descriptors. A visibility graph is a map of a network of 'visible' locations, for example, points along the cell contour. Each node of the network represents a point on the contour, and each edge represents a 'line of sight' between them. By using closeness centrality, GraVis can reliably determine whether a node along the contour is a protrusion (i.e., seen by only a few other contour points) or an indentation (i.e., seen by many other contour points). GraVis can determine these features similarly well as PaCeQuant, but only if both methods are manually parameter optimised. With default settings, GraVis performs much closer to the ground truth generated for GraVis and PaCeQuant with a group of >20 and >4 individuals, respectively. By comparing the position of lobes with the position of tri-cellular junctions, GraVis is one of the few methods that use information from neighbouring cells to obtain a better prediction accuracy of cellular shape features. However, relating shape features to intracellular entities is also lacking in this method. GraVis comes as an open-source R package and can be downloaded from GitHub as an executable file for all major operating systems (https://github.com/jnowak90/GraVisGUI).

### 5.5. LOCO-EFA: Elliptical Fourier analysis-based method (Sánchez-Corrales et al.,2018)

Elliptic Fourier analysis (EFA) decomposes segmented cell outlines into a series of ellipses, called modes or harmonics. This is comparable to conventional Fourier analysis, which decomposes a linear signal into its frequency components. In EFA, however, the deviation of the cell outline from an ellipse is used as the 'signal' and the complexity of this original signal is iteratively replaced by ellipses of increasing order that capture progressively finer details of the cell outline. The linear combination of all modes then allows the recovery of the original cell outline. The problem with this method as pointed out by Sánchez-Corrales et al. (2018) that the EFA coefficients are redundant (there are more parameters than necessary to reproduce the outline) which can render the comparison of cell shapes using principal component analyses problematic. An advancement of this method termed 'lobe contribution' EFA (LOCO-EFA) decomposes each harmonic into a clockwise and a counterclockwise rotating signal, thereby generating a unique set of parameters. This approach thus provides unambiguous measurements of cell shape, often in the form of a spectrum of directional components that make up the complex cell boundary. This method can be used to quantify the increase in complexity of cell shapes during growth. The disadvantage is that each cell is analysed on its own and thus no reference can be made between neighbouring

cells and cellular contents (e.g., microtubules). Also, the number and location of indentations and protrusions are not an output of this method. LOCO-EFA is delivered as open-source C code.

## 6. Conclusion

Several studies reveal a complex signalling mechanism involving both mechanical and chemical inputs that regulate pavement cell morphogenesis. Yet, it remains challenging to evaluate confidently events occurring during the initiation of the symmetry-breaking process, as the spatio-temporal regulation of such events cannot be accurately predicted. Many studies are focused on microtubule-based regulation of morphogenesis; it is necessary to further evaluate the role of actin using more advanced imaging and molecular approaches. It is unclear if pavement cell shape complexity correlates with growth and packing of cells in inner layers of the leaves to ensure proper tissue integrity. While several hypotheses on the functional role of interdigitated cell shape have be proposed, the presence of a wide range of pavement cell shapes across several species makes it difficult to associate such shapes with one conserved functional role. It is increasingly necessary that the evaluation of cell biological processes and phenotypes requires the development of advanced quantitative tools. While image analysis tools have received increasing attention, it is also necessary to develop molecular tools that provide a quantitative readout of the physical status of the cell. The development of fluorescence energy transfer-based mechanical stress sensors or ways to transiently perturb specific molecular effectors at different scales using optogenetic methods and track morphological changes over time would allow us to further enhance our understanding of mechanical stress-based regulation of morphogenesis in a quantitative manner.

### Acknowledgments

We thank all the reviewers for constructive comments on the review.

**Financial support.** This work was supported by the project SHAPENET, 031L0177B, of the German Federal Ministry of Education and Research grants to A.S.

**Competing interest.** The authors declare none.

**Authorship contribution.** R.v.S., R.S. and A.S. wrote the manuscript.

**Data availability statement.** Not applicable to this article as no datasets were generated or analysed here.

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
