## [Reviewer Report]

*Comments to Author*: To the Authors,

I recommend your paper for acceptance with revisions. It is a well-written review that summarises a topic which has many complexities. It is also a topic that has needed a review for quite a while now. The inclusion of a review of the different quantitative approaches to evaluate pavement cell morphology should be very beneficial to the field.

Please see below my suggestions.

The only general overarching suggestion is that I think it would be nice to better highlight the current gaps in the field and remaining questions.

My specific suggestions for improvement are.

Line 49: (But links to line 86) It is not made clear that the non-activated ROPs are not membrane-bound. This behaviour could be influential in the signalling dynamics as it means they then have different diffusion speeds and could set up possible Turing patterns. Maybe cite A.R.Champneys (2021) https://doi.org/10.1016/j.physd.2020.132735 for an example.

Line 92: The word CRIB is not defined. There are other terms here that may be unfamiliar to a reader.

Line 120: Isn’t the nano-clustering linked to local reduction in diffusion speed, but what is downstream from this? Maybe an open question to highlight?

Line 143 -Line 154: This paragraph does not truly relate to the review topic at the moment. How does this link into lobe/indent formation? This needs to be reworded.

Line 195: There should be more pectin detail here, i.e. the detail in line 213, as things seem a bit out of order, and I found it a bit jarring to go straight into the pectin models. It may also be worthwhile to say that there are high amounts of methylated pectin turnover and remodelling in the lobe regions P. Sotiriou (2018) https://doi.org/10.1111/plb.12681.

Line 213: Unclear “this” statement. Is “this” supportive of Bidhendi and Geitmann 2019 or Haas et al., 2020 as it could be linked to both.

Line 229-230: Maybe say anticlinal microtubules to ensure the reader knows you are talking about those ones.

Line 253: State positive feedback loop but have yet to mention the stress-microtubule relationship. So the reader may know that microtubules in indents→stiffening→longer indent→ stress, but not where that continues. Same for line 258.

Line 254: (relates to line 266) I suggest citing/stating more evidence for microtubule-stress interactions, as some readers may not be persuaded of this relationship.

Line 258: Unclear. Hasn’t this already been said with your comments on the positive feedback loop on line 253? Is this referencing a supposed initiation strategy? If so, make it clear.

Line 290: Make it clear that you are talking about the anticlinal 2D curvature

Line 358: Maybe make it clear that these methods can label a pavement cell with fewer lobes but bigger amplitudes as the same as another with more lobes but smaller.

Line 375: I do not think network properties is well defined here—links to line 436. I do not know if this is a clear explanation for people who do not necessarily know what a network, node or edge is. An additional diagram in figure 3 would really help.

Line 378: Not clear what the square is for in figure 3F. Also, the letter F is not referenced (it is talked about though) in the caption to figure 3

Line 418: It would be good to reference the figure here.

Line 442: Unclear. What are you referring to here by the gold standard and individuals?

Line 456: Not too sure whether finer-grained is the correct word to use to mean higher-order equations.

Line 459 and line 466: Not too sure that I agree, you can use the coefficients in PCA and get the dominant behaviour. See figure 1F in this article https://www.biorxiv.org/content/10.1101/2020.12.07.415307v1.full.pdf for example which uses this method effectively to characterise cell shape.

Line 476-480: Elaborate a bit more; this feeds into my point of there being a lack of open questions mentioned. Maybe discuss that how these shapes are initiation is still a mystery

---

## [Reviewer Report]

*Comments to Author*: In this manuscript the authors present a review of the mechanisms regulating pavement cell shape morphogenesis, focusing on some of the latest developments in the field and with a useful review of some of the computational methods used to quantitatively characterise the shapes of pavement cells.

Apart from some minor comments, suggestions and typos (see next paragraph) my main comment on the content of the text is regarding section 4 (Quantitative approaches to evaluate pavement cell morphology). The authors describe different tools including the ones they have developed themselves. While for all other tools they also mention some limitations of the approaches with a similar sentence for each tools, e.g. ”The drawback is that each cell is analyzed on its own and thus no reference can be made between neighboring cells and intracellular entities (e.g., microtubules or other proteins). Furthermore, the position of lobes and necks is not output directly and tri-cellular junctions are considered “true” lobes if not removed manually.” This is with the exception of the paragraph that describes the method they developed. To me it is not clear that their method doesn’t have most of these drawbacks as well. The only drawback that seems to be absent in this case is related to the “reference to intracellular entities” as far as I understand from this text. More generally, I think the authors also need to add a similar section listing the limitation of their method in this paragraph. They could also for instance mention the drawback that this tool is a Matlab script, Matlab being a proprietary software for which licenses are not available in every lab/institution.

Other minor comments:

- There are two sections named 2.3

- Could the authors provide a reference supporting the sentence line 282-285.

- Line 297: “to freely along certain paths” is missing the ”move” I suppose.

- Sentence line 298-300 is very difficult to understand and seems to be missing some words as well.

- Line 328: Propidium Iodide is a not a membrane specific dye. It binds to the cell wall. Maybe the authors meant to mention to FM4-64?

- Line 286-387: As far as I know the method works on 2D images, thus I do not think a user can directly start with a confocal Z-stack.

- Line 391-392: As far as I understand from the PaceQuant publication and website, PaceQuant can automatically classify lobe types (Lobe/neck vs. tricellular junction).

- Line 410: Was “fluorescence signal” meant instead of “protein signal”?

- Line 410: Could the authors clarify what is meant by ‘“true” cell surface’? Is this more “true” that with other methods? Also is the method mentioned here different form the one described in the publication “Shihavuddin A et al. Smooth 2D manifold extraction from 3D image stack. Nat Commun. 2017;8:15554.”

---

## [Reviewer Report]

*Comments to Author*: Both the Reviewers recommend minor revision of this submission. Please follow carefully their general and more specific comments, which in my opinion will improve the manuscript and make it ready for publication.

---

## [Reviewer Report]

Dear Olivier.

Please find the revised version of the review “Mechano-chemical regulation of complex cell shape formation – epidermal pavement cells a case study” for your assessment.

Best Wishes,

arun

---

## [Reviewer Report]

*Comments to Author*: In this revised version of the manuscript the authors seem to have answered most of the comments. However, I noticed that while they have corrected the text based on my comment on the capacity of PaCeQuant to distinguish between lobes and lobes associated with tricellular junction (around line 405), line 446 still mentions that PaCeQuant does not easily provides this information. More generally, I would recommend the authors to thoroughly check the consistency of their manuscript following the corrections that have been made in the revision in case similar inconsistencies remain.

I am also surprised by the new image presented at the beginning of the manuscript (Graphical abstract? Maybe I missed it in the first submission?). I do not really understand what is the image presented at the top. What is the signal? How does the displayed signal relate to mechanical stress intensity? Maybe it would be better to just draw a diagram instead of microscopy images that will miss a caption describing the actual signal displayed?

---

## [Reviewer Report]

*Comments to Author*: To the Authors,

I recommend your paper for acceptance. This a very good review that gives an up-to-date efficient summary of a complex topic. I thank the authors for making the requested changes which have improved the manuscript’s clarity.

---

## [Reviewer Report]

*Comments to Author*: The manuscript has been improved as appreciated by both the Reviewers but it still requires some minor corrections. Please follow the advice of Reviewer #1. I expect that after introducing these Reviewer’s suggestions the manuscript will be ready for publication.

---

## [Reviewer Report]

*Comments to Author*: The manuscript has been corrected as suggested by the Reviewer and is now ready for publication.